# Novel Cell Models to Study Myelin and Microglia Interactions

**DOI:** 10.3390/ijms26052179

**Published:** 2025-02-28

**Authors:** Marta Santacreu-Vilaseca, Judith Moreno-Magallon, Alba Juanes-Casado, Anna Gil-Sánchez, Cristina González-Mingot, Pascual Torres, Luis Brieva

**Affiliations:** 1Metabolic Pathophysiology Research Group, Department of Experimental Medicine, University of Lleida-IRBLleida, 25198 Lleida, Spain; marta.santacreu@udl.cat (M.S.-V.); alba.juanes@irblleida.udl.cat (A.J.-C.); 2Neuroimmunology Group, Department of Medicine, University of Lleida-IRBLleida, 25198 Lleida, Spain; jmoreno@irblleida.cat (J.M.-M.); agil@irblleida.cat (A.G.-S.); cgonzalezm.lleida.ics@gencat.cat (C.G.-M.); 3Department of Neurology, Hospital Universitari Arnau de Vilanova, 25198 Lleida, Spain

**Keywords:** multiple sclerosis, neurodegeneration, demyelination, oxidative stress, microglia, primary cell culture, personalized medicine

## Abstract

Multiple sclerosis (MS) is characterized by demyelination and neuroinflammation, with oxidative stress playing a pivotal role in lesion pathology. This study aimed to investigate the differential cellular responses to myelin debris under varying oxidative states. Myelin oxidation was induced using a Cu–peroxide system, confirmed by elevated TBARS levels and autofluorescence. BV-2 microglia viability remained unaffected by myelin exposure. However, oxidized myelin significantly altered oxidative stress markers, autophagy, and iron metabolism, as evidenced by changes in Sod2, Tfr1, p62, and P-Erk/Erk ratios. Morphological analyses revealed time- and dose-dependent differences in myelin processing, with oxidized myelin leading to distinct phagosome dynamics. Complementary studies using induced microglia-like cells (iMG)—a primary cell culture—confirmed the feasibility of employing oxidized microglia to study microglia activity. The use of iMGs provides a model closer to patient physiology, offering the potential to evaluate individual cellular responses to oxidative damage. This approach could be instrumental in identifying personalized therapeutic strategies by assessing patient-specific microglial behavior in response to myelin debris. These findings highlight the impact of myelin oxidative status on microglial function, advancing the understanding of oxidative stress in MS and paving the way for personalized medicine applications in neuroinflammation.

## 1. Introduction

Multiple Sclerosis (MS) is an autoimmune, chronic, inflammatory, and demyelinating disease of the Central Nervous System (CNS) [1].

It is characterized by an erroneous response of the immune system that targets and damages the myelin sheath, a protective layer encompassing nerve fibers in the CNS, leading to demyelination and axonal degeneration. The resultant damage disrupts the transmission of nerve impulses, giving rise to neurological symptoms including muscle weakness, balance and coordination impairments, fatigue, visual impairment, cognitive deficits, and sensory disturbances [2].

MS is one of the most prevalent neurological conditions worldwide and, in numerous countries, ranks as the foremost cause of non-traumatic neurological disability among young adults (20–40 years), affecting females approximately twice as frequently as males. Globally, it is estimated that MS afflicts 2.8 million individuals, but the prevalence varies across regions, with a prevalence of 111–300 cases per 100,000 people in the European and American regions and a lower prevalence of 5 per 100,000 people in the African and Western Pacific regions [3,4].

This disease commonly starts with periodic relapses followed by remissions, called relapsing–remitting MS (RRMS). Relapses are linked to an acute peripheral immune response targeting the CNS, which is limited in time and can be partially or fully resolved. Fortunately, in the last decades, 19 Disease-Modifying Therapies (DMT) have been approved for MS treatment, dramatically changing the lives of MS patients [3]. However, most patients with RRMS eventually shift to Secondary Progressive MS (SPMS), characterized by a gradual and continuous worsening of their condition, independent of relapses, following an initial phase of relapsing disease [5]. This transition to SPMS is mainly driven by silent progression, which is commonly known as Progression Independent of Relapse Activity (PIRA). PIRA takes place in the early stages of the disease [6] and, in contrast to the peripheral immune activation of the relapses, is driven by the innate immune system, mainly by microglia cells localized in the CNS [7]. Since most of the DMTs for RRMS target peripheral immunity, they lack efficacy in modifying SPMS progression, making this a critical gap in MS treatment.

Microglial activation is found in MS lesions associated with disease progression [8]. This activation arises as a consequence of demyelination and oligodendrocyte renewal subsequent to oxidative lesion development. The precise role of microglia in neurodegeneration remains a subject of extensive debate. Following demyelination, microglia assume crucial functions in the phagocytic elimination of myelin debris [9]. The activation of microglia leads to an increased production of reactive oxygen species (ROS) and nitric oxide, producing a potentially toxic environment for axons. Oxidative damage is amplified by the accumulation of iron, which is released by oligodendrocytes and sequestered by microglia, typically at the periphery of lesions and within deep gray matter structures [9,10].

Therefore, microglia cells are an excellent candidate to treat SPMS, but because of their location in the CNS, they cannot be collected while the patient is alive, in contrast to peripheral immune cells. Therefore, alternative methods to study these cells in the context of MS pathology are highly valuable.

In this study, we aim to generate a novel cell system based on microglia cells and their exposure to oxidized myelin debris. Within this, we can study how an MS-like environment (defined by oxidized myelin exposure) influences the two main functions of microglia: inflammation and phagocytoses.

To generate this model, we selected, on one hand, BV-2 cells (derived from C57BL/6 murine), because they are widely used as an in vitro model due to their ability to perform key microglial functions, including phagocytosis and cytokine production. They provide a reproducible and cost-effective alternative to primary microglia, which can be challenging to isolate and maintain. On the other hand, we validated the feasibility of this model in human primary cells derived from transdifferentiated circulating monocytes to induced microglia-like cells (iMG). There are several reasons why iMG use can better represent human microglia than BV-2 cells and why they are useful for MS research. First, iMG cells bear the genetic background of the donor. Therefore, we can keep these genetic factors, which might be important for the pathology. Notably, genes involved in MS risk are enriched in microglia [11]. Second, we could analyze the individualized response by differentiating iMG from MS patients and testing a battery of compounds to anticipate the response to treatment after a pro-inflammatory stimulus. Third, iMG are human origin cells, in contrast to BV-2 cells, and are not immortalized, thus preserving natural cell cycle machinery.

All in all, our results indicate that the oxidative status of myelin debris is relevant for its processing and can alter antioxidant enzyme production and autophagy in BV-2 cells. Moreover, we demonstrate that oxidized myelin debris are internalized by iMGs, showcasing the feasibility of performing an individualized analysis of the iMG response to myelin exposure [10,12,13,14,15].

## 2. Results

### 2.1. Myelin Oxidation

We aim to obtain oxidized myelin to better reproduce the lesional environment in MS and to discriminate the potential differential response due to the oxidative status. Since myelin is mainly composed of lipids, we performed a TBARS assay to quantify the malondialdehyde molecule (a relevant by-product from oxidized lipids). Oxidized myelin was obtained using a Cu-peroxide system, and it resulted in a significant increase in MDA content (measured by TBARS) (Figure 1A). Visual inspection of the myelin samples further substantiates this contrast, wherein the oxidized myelin exhibits a distinct brown coloration, compatible with a Maillard reaction, and the CTL myelin displays a whitish hue (Figure 1B). Oxidized elements include some autofluorescent compounds, like lipofuscin. Therefore, the level of autofluorescence is associated with an increase in oxidized molecules. In the case of oxidized myelin, we obtained a higher autofluorescence emission when exciting the sample at 488 nm, compared to the buffer (left tube) and CTL myelin (middle tube) (Figure 1C).

### 2.2. Cellular Viability Is Not Compromised After Myelin Exposure

We wondered whether exposure to different amounts and oxidation statuses of myelin could influence cellular viability. We performed a viability assay with Presto Blue reagent after 24 h of myelin exposure. Neither the dose nor the oxidation status changed the cellular viability of the BV-2 cell line after 24 h (Figure 2).

### 2.3. Myelin Exposure Triggers Changes in Antioxidant Enzymes, Autophagy Markers, and Proteins Regulating Iron Metabolism in BV-2 Cell Line

We aimed to analyze the changes in different pathways related to oxidative stress and cell survival as a response to myelin exposure. We took advantage of Western blot analysis to quantify changes in protein expression. Following statistical analysis using one-way ANOVA, the only proteins that demonstrated significant differences among the treatment groups were as follows: mitochondrial superoxide dismutase 2 (Sod2, *p* < 0.0001); transferrin receptor protein 1 (Tfr1, *p* < 0.01); p62, also known as sequestosome 1 (*p* < 0.01); and the ratio of phosphorylated extracellular-signal-regulated kinase to total ERK (P-Erk/Erk total, *p* < 0.05).

Regarding Sod2, statistically significant differences were observed between all treatment groups and the untreated control group, with particularly notable distinctions between the no treatment (NT) group and oxidized myelin (2 mg), and between NT versus CTL myelin (4 mg), both displaying a *p* ≤ 0.0001. All treated groups exhibited lower Sod2 levels in contrast to the NT group (Figure 3A).

In the case of Tfr1, significant differences were found in the comparisons between the NT group and the CTL myelin group (4 mg), with *p* ≤ 0.05, as well as between the NT group and the oxidized myelin group (4 mg), with *p* ≤ 0.01. Both treated groups displayed higher levels of this protein in comparison to the untreated groups (Figure 3B).

For p62, all treated groups exhibited elevated levels of this autophagy protein relative to the NT group. However, only the NT group and oxidized myelin (4 mg) comparison yielded a significant *p*-value of ≤0.01 (Figure 3C).

Finally, in relation to the P-Erk/Erk total ratio, a significant difference was observed between CTL myelin (2 mg) and oxidized myelin (4 mg), with *p* ≤ 0.05. This indicates that the former displayed a lower ratio (and thus less phosphorylated Erk), while the latter exhibited a higher ratio (indicative of more P-Erk) in comparison to the average ratio of the NT group (Figure 3D).

Figures presenting the statistical analysis for proteins that did not yield significant results can be found in (Appendix A) alongside images of the conducted quantifications (Appendix A).

### 2.4. Time-Course of Morphological Analyses of Phagocyted Myelin in BV-2 Cells

We wanted to investigate the differential capability of BV-2 cells in processing myelin depending on time, oxidated status, and dose. We tagged the myelin with a fluorescent dye (CFSE) to track it. Using the program Cell Profiler 4.2.6, multiple parameters related to intracellular phagosomes were investigated, including their quantity per cell, as well as their area, perimeter, and eccentricity. Additionally, the fluorescence intensity emitted by myelin, captured in a series of images from the wells (10 images per well), was examined. Remarkably, all these analyses yielded *p*-values < 0.0001 for both the temporal and treatment factors, as well as for their interaction.

Concerning the number of intracellular phagosomes per cell (Figure 4A), it was discerned that following a 3 h treatment, the cell population displayed less than one phagosome per cell, with higher quantities observed in cells subjected to elevated myelin doses (4 mg), particularly in the oxidized myelin group (4 mg). Analogously, at the 6 h time point, a similar pattern emerged, exhibiting a substantial increase in the number of intracellular phagosomes within the CTL myelin (4 mg) group. Subsequently, in the 24 h treatment group, a notable trend alteration ensued, with a greater abundance of intracellular phagosomes observed in groups treated with oxidized myelin, exceeding two phagosomes per cell in the case of the oxidized myelin (4 mg) regimen. Conversely, the CTL myelin groups displayed a minimal increase (relative to the 6 h one) when treated with both 2 and 4 mg myelin doses.

When evaluating the eccentricity (e) parameter to assess the shape of the phagosomes and their propensity for roundness, it was consistently observed that the eccentricity values surpassed 0.5 in all instances, indicating a lack of symmetry (Figure 4B). Specifically, during the 3 h treatment, it became apparent that lower myelin doses (2 mg) for both oxidized and CTL myelin treatments yielded similar and lower eccentricity values, whereas higher myelin doses (4 mg) resulted in elevated eccentricity, particularly notable in the oxidized myelin (4 mg) group. However, at the 6 h time point, this pattern shifted. CTL myelin treatments exhibited a considerable increase in eccentricity compared to the previous time point, predominantly observed in the CTL myelin (4 mg) group, which surpassed the eccentricity of the oxidized myelin (4 mg) group. On the other hand, treatments with oxidized myelin demonstrated a slight elevation in eccentricity compared to the preceding time point, with a more significant alteration observed in the 2 mg oxidized myelin treatment. Lastly, after a 24 h treatment, the eccentricity of phagosomes within groups subjected to higher myelin doses escalated to the point of nearly equalizing. Regarding the eccentricity resulting from the 2 mg treatments, the CTL myelin treatment exhibited a marginal increase compared to the previous time point, while the oxidized myelin treatment displayed a noteworthy elevation, almost reaching a comparable level of eccentricity.

Myelin-treated cells exhibited a greater perimeter compared to those in the CTL myelin groups at the initial time point examined (3 h). In the case of oxidized myelin, the perimeter was larger when subjected to a 4 mg dosage as opposed to a 2 mg dosage, while in the case of CTL myelin, the reverse was observed, with a larger perimeter noted in the 2 mg treatment compared to the 4 mg treatment. This trend of larger phagosome perimeters in the groups treated with oxidized myelin persisted at 6 h. However, both the CTL myelin and oxidized myelin groups displayed larger perimeters in the phagosomes resulting from treatments with a lower dosage (2 mg) compared to those with a higher dosage (4 mg) of the same myelin type (oxidized or control). Particularly intriguing is the case of oxidized myelin at 4 mg, which exhibited a reduction in perimeter when contrasted with its measurement at 3 h. When shifting our focus to the 24 h treatment, it became apparent that in the groups treated with CTL myelin, the phagosomes underwent a drastic reduction in perimeter in both instances (2 mg and 4 mg), ultimately reaching an equivalent size. In contrast, within the group treated with oxidized myelin, the phagosomes subjected to the 2 mg treatment demonstrated a slight decrease in perimeter compared to the previous time point, while those in the 4 mg treatment experienced a significant increase in perimeter (Figure 4C).

Upon examining the area (Figure 4D), two distinct trends emerged. Three of the groups (CTL myelin 2 mg, CTL myelin 4 mg, and oxidized myelin 2 mg) displayed an increase in phagosomal area from 3 to 6 h, followed by a subsequent decrease in this area from 6 to 24 h. In contrast, the area of intracellular phagosomes subjected to the oxidized myelin 4 mg treatment experienced a decrease from 3 to 6 h, subsequently followed by a significant increase from 6 to 24 h. This pattern aligns consistently with previous observations made during the analysis of perimeter (Figure 4C). Specifically, at 3 h, the phagosomal area in cells treated with oxidized myelin was practically identical, while in the case of the CTL myelin treatment, the area was greater when a lower dosage of myelin (2 mg) was administered compared to a higher dosage (4 mg). Once again, at 6 h, it was observed that in treatments involving oxidized myelin, the area consistently exceeded that of the control groups across all time points, except for the area of phagosomes in the CTL myelin 2 mg treatment, which equaled that of the phagosomes in the oxidized myelin 4 mg treatment. It is notable that at 24 h, the area of intracellular phagosomes resulting from the CTL myelin treatment (similarly observed in the perimeter analysis) underwent a dramatic reduction, nearly aligning with the area of phagosomes in the oxidized myelin group. Conversely, in the case of oxidized myelin, the area of the oxidized myelin 4 mg treatment was increased, surpassing that of the 2 mg treatment, while the latter experienced a decrease compared to the previous time point (6 h).

Finally, the total fluorescence intensity of intracellular myelin was assessed (Figure 5). The analysis revealed two distinct temporal trends based on the treatment groups. The oxidized myelin groups exhibited a consistent ascending pattern, wherein the fluorescence intensity progressively increased with longer treatment durations. In contrast, the control groups demonstrated a contrasting trend characterized by an initial increase in fluorescence from 3 to 6 h, followed by a subsequent decrease at the 24 h mark. This pattern closely resembled the observations made in the perimeter and area analyses. Notably, across all time points, the oxidized myelin 4 mg group consistently exhibited the highest fluorescence intensity.

### 2.5. Myelin Exposure Does Not Induce a Pro-Inflammatory Profile in BV-2 Cells

One of the most important features of microglia is the release of pro-inflammatory cytokines upon different noxious stimuli. After conducting a statistical analysis using one-way ANOVA on the three investigated cytokines measured by RT-qPCR, there were no statistically significant differences found among the studied cytokines (Figure 6), including tumor necrosis factor-alpha (*Tnfa*) (*p*-value 0.4245), interleukin 1 alpha (*Il1a*) (*p*-value 0.1194), and interleukin 6 (*Il6*) (*p*-value 0.2922).

### 2.6. Transdifferentiated Monocytes to iMG Phagocytes Myelin

We aimed to further validate this model, based on microglia exposure to myelin (oxidized or not), using a primary culture derived from circulating monocytes and transdifferentiated into microglia (iMG). This model is more closely related to patients and can better reproduce normal microglia than BV-2 cells, although its obtainment is more time-consuming and requires more resources. After PBMCs seeding, monocytes were attached onto a Matrigel-coated plate surface. After transdifferentiation, incubation with 2 mg of oxidized myelin tagged with CFSE for 6 h, cells were visualized through fluorescence microscopy. Polarized cells compatible with microglia morphology were observed, with myelin incorporated in intracellular granules. Compared to BV-2 cells, the myelin granules seemed more diffuse and in higher numbers (Figure 7).

## 3. Discussion

In this study, we aimed to elucidate the effects of oxidized myelin on cellular processes in microglial models, using both the BV-2 cell line and transdifferentiated monocytes to induce microglia (iMG). Our findings provide insights into the oxidative status of myelin and its impact on cellular pathways related to oxidative stress, phagocytosis, and inflammatory responses.

### 3.1. Oxidative Modification of Myelin

The successful oxidation of myelin, as confirmed by TBARS assay, distinct coloration, and increased autofluorescence, underscores its utility in mimicking the lesional environment in multiple sclerosis (MS). The significant increase in malondialdehyde (MDA) content in oxidized myelin aligns with previous studies indicating lipid peroxidation as a hallmark of oxidative stress in neurodegenerative conditions. The observed autofluorescence, likely due to compounds such as lipofuscin, further corroborates the oxidative modifications. These findings establish oxidized myelin exposure as a means to reproduce lesional environments in demyelinating and neurodegenerative diseases, including MS [16,17,18,19].

### 3.2. Cellular Viability and Inflammatory Profile

Our results demonstrate that neither the dose nor the oxidative status of myelin significantly altered BV-2 cell viability after 24 h. This suggests that microglial survival is not compromised under these conditions, enabling us to investigate downstream effects without confounding factors related to cell death. Furthermore, the lack of significant changes in pro-inflammatory cytokine expression (*Tnfa*, *Il1a*, and *Il6*) indicates that myelin exposure alone does not elicit a pro-inflammatory response in BV-2 cells. These findings contrast with the pro-inflammatory milieu observed in MS lesions [9], suggesting that additional factors, such as other immune cells or extracellular signals, may be required to induce such responses. In this regard, in a more complex model employing myelinating cell culture (in which neurons, astroglia, microglia, and oligodendroglia are co-cultures), the addition of hemin to the cell media triggers myelin oxidation and ferroptosis. In this work, microglia are more susceptible to cell death than astrocytes and neurons, maybe due to the internalization of oxidated myelin. Interestingly, they found that the cytotoxic effects of hemin treatment were microglia-independent, although these cells secreted several pro-inflammatory cytokines into the cell medium [20]. Compared with our model, hemin addition to the cell media can also mediate the membrane composition of the cells independently of myelin oxidation. We also can track myelin phagocytosis by the addition of a fluorescence tag to study phagosome dynamics. The differences in cytokine release reinforce the potential interaction of cells in culture and may underscore the lack of cytokine secretion in BV-2 cells.

### 3.3. Oxidized Myelin and Cellular Pathways

Western blot analyses revealed notable alterations in oxidative stress markers, autophagy-related proteins, and iron metabolism regulators upon myelin exposure. Reduced levels of Sod2 in treated groups suggest a potential impairment in mitochondrial antioxidant defenses, which may be exacerbated in the presence of oxidized myelin. The upregulation of Tfr1 in treated groups highlights the role of iron metabolism in microglial responses, consistent with the known association between iron dysregulation and MS pathology. Increased levels of p62 further implicate autophagy in the processing of myelin debris, particularly oxidized myelin. These findings indicate that oxidized myelin influences key cellular pathways related to oxidative stress [21], iron homeostasis, potentially ferroptosis [22], and autophagy [23], which may contribute to microglial dysfunction in MS. Notably, iron accumulation is present in chronic lesions in MS patients and drives chronic tissue damage [24]. Iron accumulation can trigger ferroptosis, and there is an intriguing connection between this cell death and autophagy. Insufficient autophagy of membrane organelles can facilitate the accumulation of oxidized lipids, which in turn are generated by iron-overload-mediated ROS [25]. A downregulation of Sod2 can also compromise the antioxidant capacity of the cell, allowing widespread ROS damage. This harmful situation might lead to ferroptosis or even a stress-induced senescence and a pro-inflammatory phenotype of microglia [26]. In this regard, degenerating microglia in white matter (linked to impaired remyelination) contain higher levels of lipid peroxidation and increased expression of ferroptosis-related genes involved in iron-mediated lipid dysmetabolism and oxidative stress associated with neurodegeneration in Alzheimer’s disease and vascular dementia [27]. Ferroptosis inhibitors are being developed and proposed as potential therapies for MS [28].

### 3.4. Enhanced Phagocytosis of Oxidized Myelin

Our phagocytosis assays revealed significant differences in the processing of oxidized versus CTL myelin over time. BV-2 cells exhibited a greater number of intracellular phagosomes, as well as increased fluorescence intensity, perimeter, and area of phagosomes, particularly in response to oxidized myelin. These findings suggest that oxidized myelin is more readily internalized and processed by microglia. Interestingly, the changes in phagosome eccentricity indicate structural alterations, which may reflect differences in the dynamics of phagosome maturation and cargo degradation. The increased fluorescence intensity in oxidized myelin-treated groups further supports its prolonged retention and metabolic activity within microglial cells. In the context of MS pathology, lipid oxidation is present in the lesions and contributes to disease progression and neurodegeneration [21]. Notably, oxidized phosphatidylcholine (found in MS lesions, derived from demyelination) is a potent neurotoxin and microglia phagocytosis can prevent its harmful effects in neurons. The fact that oxidized myelin is a more potent stimulus for phagocytosis is also present in vivo in aging models [29] and might reflect the need to remove neurotoxins from the environment to prevent neurodegeneration. An insufficient capacity to remove oxidized myelin by autophagy impairment or overproduction could be the basis of SPMS and a valuable therapeutic target that can be addressed with the present microglial models.

### 3.5. Insights from the iMG Model

The use of transdifferentiated monocytes (iMG) provided additional validation of our findings in a model more representative of human microglia. The higher number and diffuse distribution of intracellular myelin granules in iMG cells, compared to BV-2 cells, suggest enhanced phagocytic capabilities in the iMG model. These observations align with the physiological role of microglia in clearing myelin debris and underscore the importance of using primary cell-based models to obtain an individualized response to myelin processing. Previous studies using this model showed that these cells recapitulate fundamental alterations in ALS, including a shift toward a pro-inflammatory profile and protein aggregation associated with the severity of each patient [30]. In our case, we added the oxidized myelin treatment for the first time, and we postulate this model as a source of information on microglial activity that might be also associated with the disease progression of MS patients. Compared to BV-2 cells, iMG are morphologically more complex, exhibiting several ramifications. Moreover, iMG polarization in some individuals resembles the naturally occurring bipolar/rod-shaped microglia [30], reinforcing the importance of keeping information on the genetic background to reproduce key aspects of the pathology. These rod-shaped microglia are associated with a rapid pro-inflammatory conversion, which can be interesting to modulate in vitro [31]. Interestingly, there are some similarities with BV-2 cells in microglial behavior, like LPS stimulation and cytokine secretion, phagocytic activity, and plasticity to adapt an active or resting phenotype [32,33].

### 3.6. Implications and Future Directions

The distinct cellular responses to oxidized myelin observed in this study have several implications for understanding MS pathology. The enhanced phagocytosis and altered oxidative stress markers suggest that oxidized myelin may contribute to microglial activation and dysfunction, potentially exacerbating neuroinflammation and neurodegeneration. Additionally, the lack of a robust pro-inflammatory response in BV-2 cells highlights the need to investigate the interplay between microglia and other cell types in the MS environment. Future studies could explore the impact of oxidized myelin on other microglial functions, such as cytokine secretion, synaptic pruning, and neurotrophic support.

In conclusion, our study provides a comprehensive analysis of the effects of oxidized myelin on microglial models, revealing key alterations in oxidative stress, autophagy, and phagocytosis. These findings contribute to our understanding of microglial responses to myelin debris in MS and pave the way for future investigations into individualized responses to oxidized myelin exposure by mimicking the lesional environment in MS. Future studies will evaluate the concordance between the response to a given treatment, in terms of clinical manifestations, and the response to the same treatment in iMGs by modulating inflammation and phagocytic activity. If we were to find some biomarker of response to treatment in those cells, we might anticipate the efficacy of it in patients before starting the therapy; thus, we could do a screening experiment seeking the best candidate for each patient based on iMG response.

## 4. Materials and Methods

### 4.1. Extraction and Oxidation of Myelin Debris

The myelin debris was extracted according to the protocol described in [34], with some modifications. Tris·Cl buffer solution was prepared by adding 20 mL of 1 M Tris·Cl and 20 mL of 100 mM Na_2_EDTA to 800 mL of distilled deionized water (ddH2O). The pH was then adjusted to 7.45, and the total volume was brought up to 1 L. This was utilized to prepare two sucrose solutions: one with a concentration of 0.32 M and the other with a concentration of 0.83 M.

The next step was euthanizing six mice, extracting their brains, and submerging them in 30 mL of the 0.32 M sucrose solution. Using a sterile hand-held rotary homogenizer, the brains were thoroughly homogenized. The homogenized brain solution was brought to a final volume of 90 mL by adding the 0.32 M sucrose solution. Ultracentrifuge tubes were loaded with 20 μL mL of the 0.83 M sucrose solution, onto which the homogenized brain solution was gently layered to create a density gradient. These tubes were then subjected to a centrifugation process, set at 100,000× *g* for 45 min at a constant temperature of 4 °C, employing minimal acceleration and deceleration. As a result, the myelin formed a visibly distinct whitish interface, which was collected and transferred to a 50 mL tube. The volume was adjusted to 35 mL using Tris·Cl buffer to ensure thorough and effective mixing. The suspension was homogenized once again. The centrifugation process was repeated, this time utilizing the maximum acceleration and deceleration settings, to effectively precipitate the myelin debris. The supernatant was discarded, leaving behind a pellet of myelin debris. This pellet was then resuspended in 10–15 mL of Tris·Cl solution. To further enhance the purification, the myelin was subjected to another round of centrifugation under the exact same conditions, resulting in a pellet that was subsequently resuspended in 5–6 mL of sterile HBSS, ensuring optimal purity.

The myelin debris suspension was divided among several pre-weighed 1.5 mL microcentrifuge tubes. These tubes were then subjected to centrifugation at 22,000× *g* for 10 min at 4 °C. After the completion of centrifugation, the supernatant was removed, and the myelin pellet was weighed. To create a control sample consisting of CTL myelin debris, the pellet from half of the tubes was diluted to a concentration of 100 mg/mL using HBSS solution. On the other hand, to generate oxidized myelin debris, we modified an existing protocol [35], and the remaining tubes were resuspended using 50 μL of CuH (800 μM), CuSO_4_·6H_2_O, and 20 mM H_2_O_2_. Following an incubation period of 16 h at 37 °C, the tubes were subjected to another round of centrifugation at 22,000× *g* for 10 min at 4 °C. The resulting pellet was then resuspended using HBSS solution to achieve a final concentration of 100 mg/mL.

### 4.2. Labeling of Myelin Debris

The fluorescent dye carboxyfluorescein succinimidyl ester (CFSE) was used to label the myelin debris [34]. This dye has a non-cytotoxic nature, allowing for the monitoring of myelin debris internalization by the BV-2 cells following treatment. Furthermore, CFSE exhibits a narrow fluorescent spectrum, facilitating its simultaneous use with other fluorescence assays. For labelling, a 50 µM CFSE solution was prepared immediately before use, using HBSS to dilute the stock solution.

The myelin debris, previously obtained at a concentration of 100 mg/mL (oxidized and CTL), was transferred to a pre-weighed 1.5 mL microcentrifuge tube. The tube was then centrifuged for 10 min at 4 °C and 14,800× *g*. After discarding the supernatant, the resulting pellet containing the myelin debris was resuspended using 200 μL of CFSE solution for every 100 μL of pellet. The mixture was incubated for 30 min at room temperature, protected from light exposure. Following this incubation period, the tube was once again centrifuged under the same conditions, and the supernatant was discarded once again. To cleanse the labeled debris, the pellet was resuspended in 600–800 μL of a wash buffer (100 mM glycine in HBSS). This centrifugation and washing process was repeated twice more. Finally, the weight of the myelin debris pellet was determined, and it was resuspended at a concentration of 100 mg/mL using sterile HBSS.

### 4.3. Cell Cultures and Administration of Treaments

For this study, a BV-2 cell line was used. BV-2 cells are an immortalized cell line derived from microglial cells obtained from C57/BL6 mice. This cell line, commercially obtained from AcceGen Biotech (Fairfield, NJ, USA, Cat# ABC-TC212S), exhibits a remarkable capacity for proliferation and metabolic activity, surpassing that of other microglia. Additionally, like other microglia, this cell line can be activated to release pro-inflammatory cytokines when exposed to inflammatory factors or oxidative stress. Consequently, the BV-2 cell line serves as an outstanding alternative model system for investigating primary microglia and studying neurodegenerative diseases in vitro [36].

Induced microglia-like cells (iMGs) were obtained as previously described [32,37]. Briefly, PBMCs were extracted using a Ficoll density gradient (Merck, Darmstadt, Germany, Cat# 10771) from one subject without clinical symptoms. Isolated PBMCs were seeded onto a pre-coated 12-well plate with 2% Geltrex (1 M PBMCs per well) in RPMI-1640 10% FBS for 1 day. The following day, the medium was replaced with differentiation medium containing RMPI-1640, 0.1 μg/mL IL-34, and 0.01 μg/mL GM-CSF for 14 days, with media changes every 2–3 days.

The BV-2 cell line was cultured in culture dishes with DMEM high glucose medium supplemented with L-glutamine, pyruvate, 10% fetal bovine serum, and antibiotic–antimycotic. Passages were performed every 2–3 days when confluence reached 70–80%. The incubator maintained a temperature of 37 °C and a CO_2_ concentration of 5%. For the treatments, 250,000 cells were seeded per well (in 1 mL of medium) in four 12-well plates with twelve wells each, using the same medium. The cells were allowed to grow for 2–3 days until reaching a confluence of 70–80%. Before applying the treatment, the medium was changed to DMEM high glucose medium supplemented with L-glutamine, pyruvate, and antibiotic–antimycotic (without fetal bovine serum). If the myelin debris had been stored at −80 °C, it was resuspended 4–5 times using a 0.5 × 16 mm needle and vigorously agitated before adding it to the cells.

Three replicates with 2 or 4 mg/mL of myelin debris (whether oxidized or CTL) were employed. These plates were used for viability assays, protein quantification, and the study of pro-inflammatory cytokines. The labeled myelin debris was applied at different time points for each row of wells (3, 6, and 24 h).

### 4.4. Viability Assay

The viability study was conducted using PrestoBlue reagent (Thermo Fisher Scientific, Waltham, MA, USA, Cat# P50200). PrestoBlue is a viability indicator that leverages the reducing capacity of living cell metabolism to convert the non-fluorescent dye resazurin (blue color) into the fluorescent molecule resorufin (pink color). To prepare the PrestoBlue solution, 1 mL of the reagent was diluted in 12 mL of DMEM high glucose medium supplemented with L-glutamine, pyruvate, and antibiotic–antimycotic. The medium in the plate was replaced with this solution, and then it was incubated for one hour in an incubator. After incubation, the fluorescence was measured using a fluorescence-based microplate reader. Nine readings were taken for each well, and the average of these readings was calculated for further statistical analysis. The obtained averages were normalized using the average fluorescence of the untreated wells and thus expressed as %NT (normalized to control). For statistical analysis, a one-way ANOVA (Tukey’s test for multiple comparisons) was performed using GraphPad Prism 8.0.2 software. This entire process was repeated four times to ensure the robustness and reliability of the results, and a *p*-value < 0.05 was considered significant.

### 4.5. Protein Expression

Firstly, the cells were harvested from the culture plate using a lysis buffer composed of RIPA supplemented with protease inhibitors (Thermo Fisher Scientific, Waltham, MA, USA, Cat# 78429), 1mM Na_3_VO_4_, and 1mM NaF to inhibit phosphatases.

The cells were initially washed with PBS, followed by the addition of 100 μL of the lysis buffer. Using a cell scraper, the adherent cells were gently detached and collected in 1.5 mL tubes, placed on ice, and sonicated. After extraction, the protein content in each sample was quantified using the Bradford protein assay, following manufacturer’s instructions (Bio-Rad, Hercules, CA, USA, Cat# 5000006). A total of 15 μg of protein were loaded into SurePAGE^TM^ precast gels (Bis-Tris, 10 × 8, 4–12%, 15 wells, Genscript, Piscataway, NJ, USA) and Tris-MOPS-SDS running buffer was employed for electrophoretic separation.

Following electrophoresis, the proteins were transferred onto PVDF membranes employing an eBlot^TM^ Protein Transfer System (Genscript, Piscataway, NJ, USA). The membranes were blocked with IBlock (Thermo Fisher Scientific, Waltham, MA, USA, Cat# T2015) and incubated overnight with the primary antibodies (Table 1) in TBS-T buffer, followed by three washing steps with TBS-T and an incubation with secondary antibodies: anti-Rabbit IgG (Thermo Fisher Scientific, Waltham, MA, USA, Cat# 31460), at dilution 1:40,000 or anti-Mouse IgG (Merck, Darmstadt, Germany, Cat# NA931) at dilution 1:40,000. After four washes with TBS-T, the membranes were revealed using Immobilon ECL Ultra Western HRP Substrate (Merck, Darmstadt, Germany, Cat# WBULS0100). Images were captured using the ChemiDoc XRS+ System (Bio-Rad, Hercules, CA, USA) and subsequently stained with Coomassie membrane stain. Band intensities were quantified using the ImageLab 6.1 software (Bio-Rad, Hercules, CA, USA), and the values were normalized based on the protein expression obtained from the membrane stain. The normalized values were expressed as %NT (normalized to control), using the average of the NT values as the reference, set at 100%. Statistical analyses were performed using one-way ANOVA (Tukey’s test for multiple comparisons) in the GraphPad Prism 8.0.2 software. The entire process was repeated four times to ensure the robustness and reliability of the results.

### 4.6. Study of Pro-Inflammatory Cytokines

After treating the cells, two washes were performed using HBSS, followed by the addition of 500 μL of TRIReagent (MRC, Montgomery Road Cincinnati, OH, USA, Cat# TR 118), and then they were collected in 2 mL tubes. A total of 100 μL of chloroform was then added. After agitation, the mixture was left to settle at room temperature for five minutes and then centrifuged at 13,000× *g* at 4 °C for 15 min. RNA was separated in the upper (aqueous) phase. The upper phase was carefully collected and transferred to 1.5 mL tubes, to which 250 μL of isopropanol was added. After vortexing, the mixture was left to rest for 10 min at room temperature and then centrifuged at 13,000× *g* for ten minutes at 4 °C. The purpose of isopropanol is to precipitate the RNA; therefore, the supernatant was discarded. Next, 500 μL of 75% ethanol (in RNAse-free water) was added to the pellet, followed by vortexing and centrifugation for five minutes at 13,000× *g* at 4 °C. The ethanol was carefully removed, and after allowing it to evaporate for 5 min at RT, the RNA was resuspended in 20 μL of RNAse-free water. Once the RNA was isolated, a Nanodrop device was used to quantify the RNA content in each sample. The samples were then diluted to a concentration of 100 ng/μL in a final volume of 10 μL using RNAse-free water.

The expression of pro-inflammatory cytokines (*Tnfa*, *Il1a*, and *Il6*) and actin as a reference gene was studied. Reverse transcription PCR (RT-PCR) was performed in order to obtain cDNA (Thermo Fisher Scientific, Waltham, MA, USA, Cat# N8080234). This cDNA was then used in quantitative PCR (qPCR) using SYBR Green. The SYBR Green dye binds to the double-stranded DNA, increasing its fluorescence emission and allowing for quantification. Each sample was analyzed (with duplicates) on a 96-well plate. The results obtained from the qPCR were analyzed using the qPCR analysis system (CFX96, Bio-Rad). The cycle threshold (Ct) values were obtained, which represent the number of amplification cycles required for the fluorescence signal to reach a detectable threshold. The relative expression of the target genes (*Tnfa*, *Il1a*, and *Il6*) was calculated using the 2^−ΔΔCt^ formula, where ΔΔCt is the difference between the Ct values of the target gene and the reference gene, actin, normalized by the average Ct in the NT condition. This expression was presented as %NT, with 100% representing the average of the NT values. Statistical analysis was performed using one-way ANOVA (Tukey’s test for multiple comparisons) with GraphPad Prism 8.0.2 software. This entire process was repeated a total of three times.

### 4.7. CFSE Assay

The cells were treated during the indicated schemes and then fixed by adding 1 mL of 4% paraformaldehyde in PBS to each well and incubating at room temperature for 30 min. Since CFSE allows simultaneous staining with other dyes, a staining with DAPI was performed. The cells were washed 1–3 times with PBS and then covered with a sufficient amount of DAPI staining solution (Thermo Fisher Scientific, Waltham, MA, USA, Cat# R37606). After incubating for 5 min, protected from light, the cells were washed again with PBS.

Next, the cells were visualized using a fluorescence microscope. Considering the excitation/emission wavelengths of 358/461 nm for DAPI and 492/517 nm for CFSE, images were taken at ten distinct positions within each well, capturing three images for each position: DAPI fluorescence, CFSE fluorescence, and a phase-contrast image. The NT wells from the other procedures were also observed to ensure there was no fluorescence in them. The obtained images were analyzed using the Cell Profiler v4.2.6 software. A pipeline was created, utilizing the nuclei detected by DAPI fluorescence to locate the cells. Phagosomes were also identified, and the RelateObjects module was used to identify intracellular phagosomes. Modules were developed to count these intracellular phagosomes, measure their area, perimeter, and eccentricity, and analyze the fluorescence intensity of CFSE-labeled myelin. This provides results on a per-cell basis.

For the myelin intensity analysis, ten values corresponding to the average intensity of each CFSE image were obtained for each condition and time point. The ten replicates (per condition and time) were statistically analyzed using a two-way ANOVA. Regarding perimeter and area measurements, the values were initially in pixels. To convert them to nanometers (nm), an image with a known scale in nm, captured under the same microscope and objective conditions, was used. This conversion was applied to both perimeter and area values. Given the large amount of data for area, perimeter, eccentricity, and intracellular phagosome counts, GraphPad Prism 8.0.2 software was employed to obtain the sample size, mean, and standard deviation for each condition and time point. Statistical analysis was performed using a two-way ANOVA (Tukey’s test for multiple comparisons) with GraphPad Prism 8.0.2 software.

### 4.8. TBARS Assay

To evaluate the success of myelin debris oxidation, a thiobarbituric acid reactive substances assay (TBARS assay) was conducted. This assay is based on the principle that peroxidation processes generate various byproducts, including malondialdehyde (MDA), as secondary products. MDA reacts with thiobarbituric acid (TBA) to form MDA-TBA2, a conjugate that exhibits absorption in the visible spectrum at 532 nm. This results in a distinctive rosy coloration, serving as a reliable indicator of the oxidation levels present in the sample.

The TBARS assay was performed following the specific protocol outlined in [20]. First, the reagents were prepared. A 3.5 M sodium acetate buffer was prepared by diluting 200 mL of glacial acetic acid in 350 mL of ddH2O. Separately, 50 mL of a 6.5 M NaOH solution was prepared, and 46 mL of this solution was slowly added to the acetic acid while continuously mixing it. The pH was adjusted to 4, and the solution was brought to a total volume of 500 mL with ddH_2_O.

The other reagents prepared were 4 mL of an 8.1% sodium dodecyl sulfate (SDS) solution in ddH_2_O, 20 mL of a 5 M NaOH solution, and 100 mL of a 0.8% aqueous solution of TBA. To ensure smooth dissolution of the TBA in the last solution, 600 μL of the 5 M NaOH solution was gradually added in 100 μL increments until the TBA was completely dissolved. The pH was then adjusted to be below 4. To prepare the MDA standards, 9.2 μL of tetramethoxypropane malonaldehyde bis was added to 100 mL of ddH2O, resulting in a final concentration of 550 μM. From this solution, standards at concentrations of 200, 100, 80, 40, 20, 10, 5, and 2.5 μM were prepared.

For the TBARS assay, 25 μL of the standard solutions were added to 2 mL tubes. For the samples, 12.5 μL of oxidized and CTL myelin debris, as well as 12.5 μL of ddH_2_O, were added to 2 mL tubes. To each tube, 50 μL of the 8.1% SDS solution, 375 μL of the 3.5 M sodium acetate buffer, and 375 μL of the 0.8% TBA solution were added. The tubes were vortexed and incubated for 1 h at 95 °C. Afterward, the tubes were cooled for 30 min on ice and centrifuged at 4000× *g* for 10 min at 4 °C. A total of 150 μL of the supernatant from each sample/standard was transferred to separate wells in a 96-well plate (three wells per sample). A spectrophotometer was utilized to measure the absorbance at 532 nm. A standard curve was created using the absorbance values of the standards, and its equation was used to extrapolate the MDA concentration in the myelin debris samples.

The concentrations of MDA obtained from the three replicates of both the control and oxidized myelin debris were subjected to statistical analysis using an unpaired Student’s *t*-test conducted with GraphPad Prism 8.0.2. This analysis allowed for the comparison and assessment of significance between the groups.

## Figures and Tables

**Figure 1 ijms-26-02179-f001:**
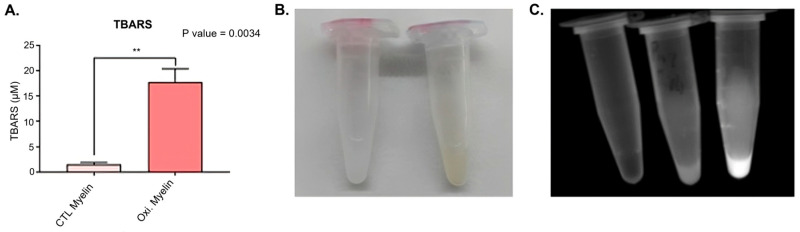
The oxidation measurements in myelin debris. (**A**) The TBARS assay outcomes: there exist statistical differences in oxidative levels between the control and oxidized myelin. (**B**) An image depicting the CTL myelin samples (on the left) and oxidized myelin samples (on the right) at a concentration of 100 μg/μL. (**C**) An image capturing the autofluorescence emitted by the samples, arranged from left to right: buffer, CTL myelin, and oxidized myelin. CTL = Control condition. Oxi. = Oxidized. ** indicates *p* < 0.01. TBARS = Thiobarbituric acid reactive substances.

**Figure 2 ijms-26-02179-f002:**
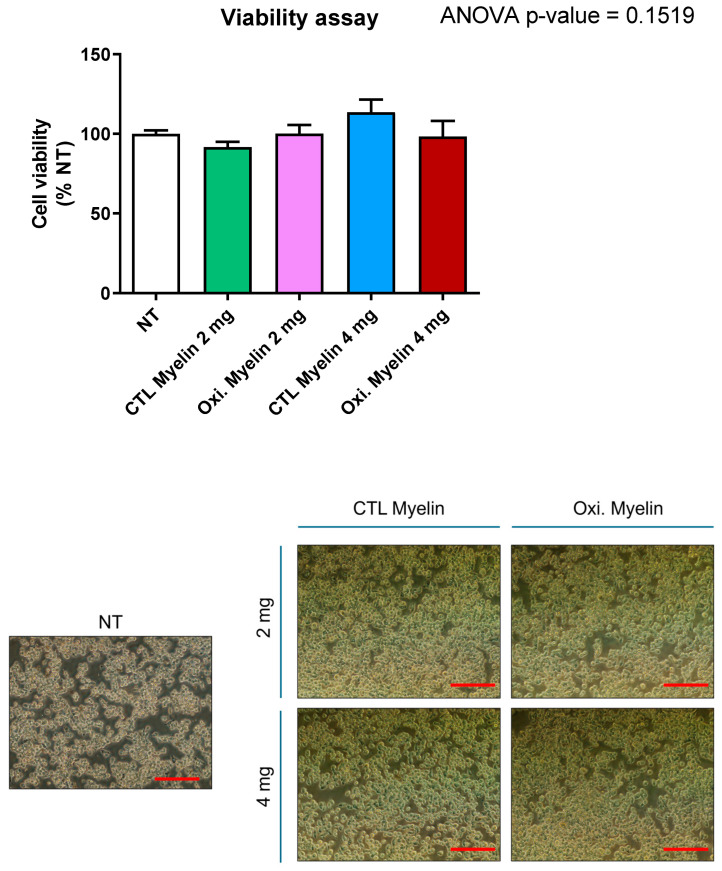
BV-2 viability after myelin exposure. Cell viability assay in BV-2: Myelin debris exposure does not alter cell viability, independently of dose and oxidation state. NT = Not treated. CTL = Control condition. Oxi. = Oxidized. Ns = Non-significant. Red scale bar indicates 200 μm.

**Figure 3 ijms-26-02179-f003:**
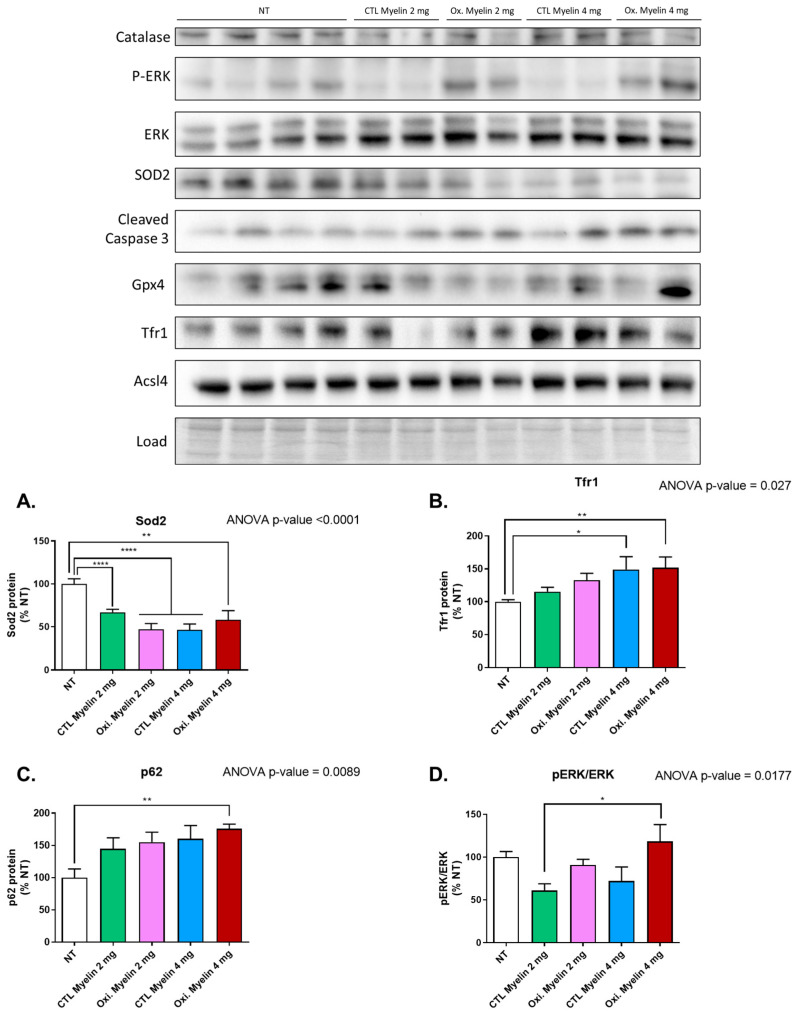
Densitometric analyses of proteins with statistically significant changes in BV-2 after myelin exposure. Results for statistically significant protein analyzed. (**A**) Superoxide dismutase 2 (Sod2), (**B**) transferrin receptor protein 1 (Tfr1), (**C**) p62, and (**D**) phosphorylated extracellular-signal-regulated kinase to total ERK ratio (P-Erk/Erk total). NT = Not treated. CTL = Control condition. Oxi. = Oxidized. *, **, and **** indicate, respectively, *p* < 0.05, *p* < 0.01, *p* < 0.0001.

**Figure 4 ijms-26-02179-f004:**
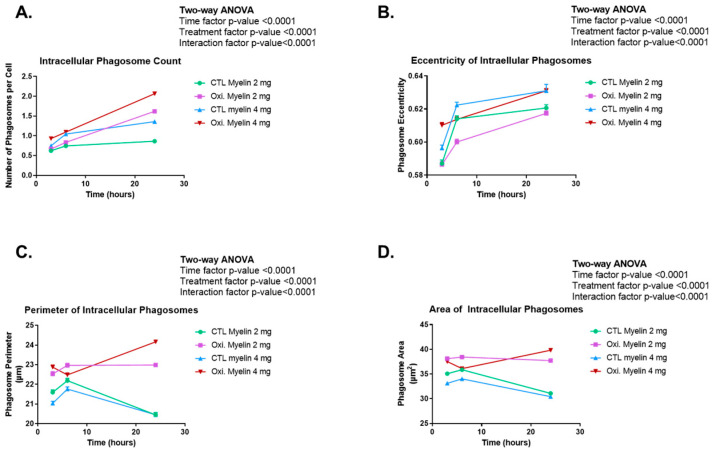
A morphological analysis of phagocytosis data in BV-2 cells. The results obtained from the CFSE assay indicate that both oxidation status and myelin dose influence the number of phagosomes, and cells treated with oxidized myelin exhibited a higher number of phagosomes compared to control myelin-treated cells in both doses (**A**). Eccentricity (a measure of circularity, in which a value of 0 is a perfect circle) increased in time in all conditions, especially for the highest dose of myelin (**B**). The perimeter of phagosomes (**C**) and the area (**D**) decreased in time for control myelin but not for oxidized myelin. CTL = Control condition. Oxi. = Oxidized.

**Figure 5 ijms-26-02179-f005:**
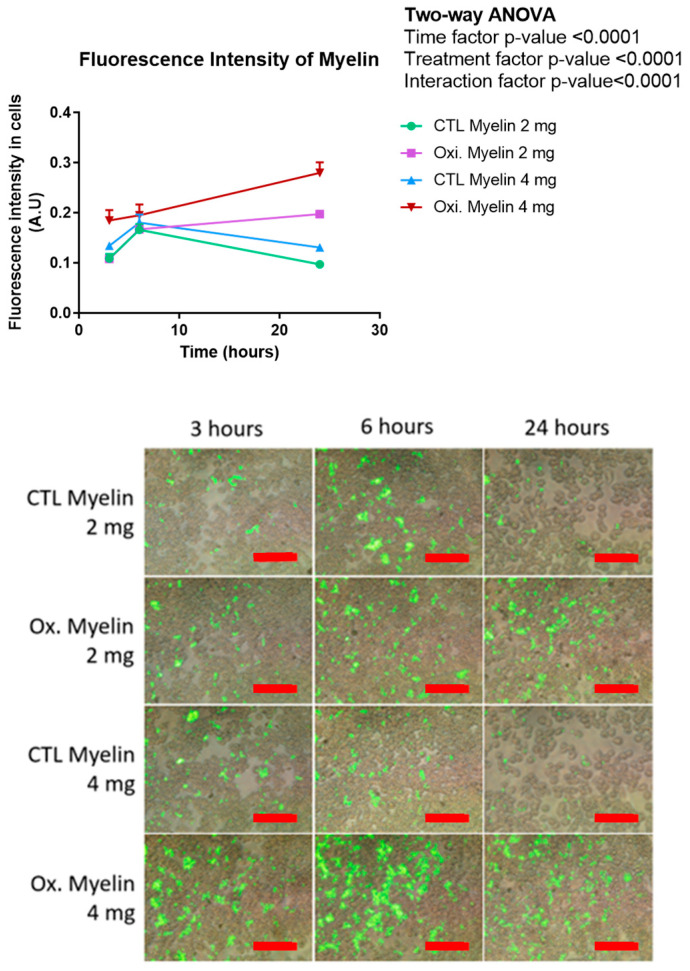
Fluorometric quantification of phagocytosis data in BV-2 cells. Fluorescence intensity of different treatments CFSE-myelin indicates higher persistence of oxidized myelin in cells compared with control myelin-treated cells. CTL = Control condition. Oxi. = oxidized. Red scale bar indicates 200 μm. A.U = Arbitrary units.

**Figure 6 ijms-26-02179-f006:**
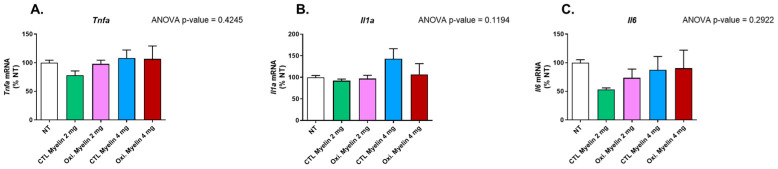
Gene expression analyses of pro-inflammatory cytokines in BV-2 cells. Gene expression of pro-inflammatory cytokines as percentage relative to mean of untreated (NT) groups. (**A**) Tumor necrosis factor-alpha (*Tnfa*), (**B**) interleukin 1 a (*Il1a*), and (**C**) interleukin 6 (*Il6*). CTL = Control condition. Oxi. = Oxidized. NT = Not treated.

**Figure 7 ijms-26-02179-f007:**
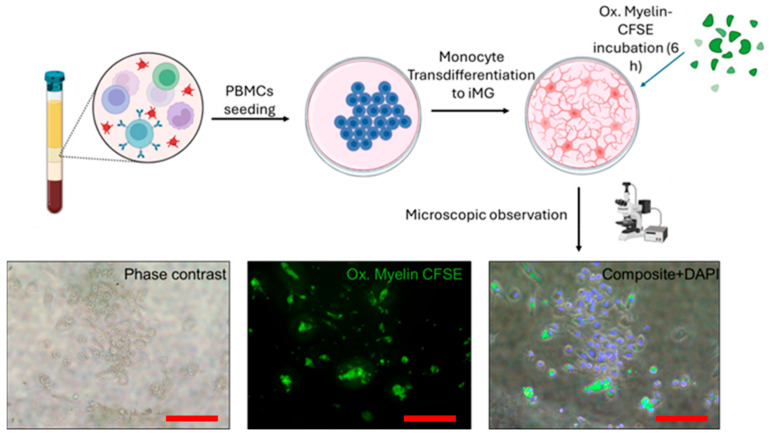
Myelin phagocytosis of iMG cells. After PBMCs extraction and monocyte transdifferentiation, resulting induced microglia (iMG) can phagocytose oxidized myelin debris tagged with CFSE. Ox. = Oxidized. iMG = induced microglia-like cells. CFSE = Carboxyfluorescein succinimidyl ester. PBMCs = Peripheral Blood Mononuclear Cells. DAPI = 4′,6-diamidino-2-phenylindole. Red scale bar indicates 100 μm.

**Table 1 ijms-26-02179-t001:** Antibodies used for protein expression using Western blot method.

Protein	Dilution	Reference	Secondary Ab	kDa
Sod2	1:1000	ab2787145 (Abcam, Cambridge, UK)	Mouse	20
P-Erk	1:1000	ab2809161 (Abcam, Cambridge, UK)	Rabbit	42–44
Erk ½ total	1:1000	ab184699 (Abcam, Cambridge, UK)	Rabbit	42–44
Catalase	1:1000	ab16731 (Abcam, Cambridge, UK)	Rabbit	60
Cleaved caspase 3	1:1000	9661(Cell signaling, Danvers, MA, USA)	Rabbit	32
Gpx4	1:1000	67763-1-(ProteinTech, Rosemont, IL, USA)	Mouse	20
Tfr1	1:500	13-6800 (Thermo Fisher Scientific, Waltham, MA, USA)	Mouse	84
Acsl4	1:1000	ab155282 (Abcam, Cambridge, UK)	Rabbit	75
p62	1:1000	5114 (Cell signaling, Danvers, MA, USA)	Rabbit	62
Lc3b	1:1000	2775 (Cell signaling, Danvers, MA, USA)	Rabbit	14–18

## Data Availability

Data are available on reasonable request.

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
