# Peer review of "Novel Cell Models to Study Myelin and Microglia Interactions"

_ijms, 2025, doi:10.3390/ijms26052179_

Round 1

Reviewer 1 Report

Comments and Suggestions for Authors

The article entitled “Novel cell models to study myelin and microglia interactions,” want to show impact of myelin oxidative status on microglial function, advancing the understanding of oxidative stress in MS and paving the way for personalized medicine applications in neuroinflammation. However, there still some main points for improvement or further consideration. 

  1. The introduction lacks a clear and logical flow, making it difficult to understand the research context.
  2. Cellular viabilityof BV-2 cell line y is not compromised after myelin exposure. The Cell morphology of BV-2 cell after myelin exposure is suggested to present with images. .
  3. Thestatistical analysis for Western blot data are present. However, the representative bands for Western blot are suggested to present.
  4. Change 'western blot' to 'Western blot'.
  5. In figure 6 and 7, there should be scale bar for the images.
  6. Define all acronyms in figure legends.

Author Response

Comments and Suggestions for Authors

The article entitled “Novel cell models to study myelin and microglia interactions,” want to show impact of myelin oxidative status on microglial function, advancing the understanding of oxidative stress in MS and paving the way for personalized medicine applications in neuroinflammation. However, there still some main points for improvement or further consideration. 

  1. The introduction lacks a clear and logical flow, making it difficult to understand the research context.

Response: Thank you for pointing this out. We agree with your comment. Therefore, we have deeply modified this section to make it more easily to read, following a logical flow.

  1. Cellular viability of BV-2 cell line y is not compromised after myelin exposure. The Cell morphology of BV-2 cell after myelin exposure is suggested to present with images.

Response: Thank you for your thoughtful observation. We present representing images from the conditions exposed in the cell viability assay figure.

  1. The statistical analysis for Western blot data are present. However, the representative bands for Western blot are suggested to present.

Response: Thank you for your thoughtful observation. We move the representative bands for Western Blot from the Figure S1 to the Figure 3 in the main text in the revised manuscript.

  1. Change 'western blot' to 'Western blot'.

Response: Agree. We modified it in the revised manuscript.

  1. In figure 6 and 7, there should be scale bar for the images.

Response: Thank you for pointing this out. We add a scale bar to all the figures from microscopic analysis.

  1. Define all acronyms in figure legends.

Response: We appreciate your careful review and valuable feedback on this matter. We added this information in all Figure Legends in the revised manuscript.

Reviewer 2 Report

Comments and Suggestions for Authors

The manuscript presents an interesting study on microglia-myelin interactions, particularly in the context of oxidative stress and multiple sclerosis (MS). The experimental design is well thought out, and the findings contribute valuable insights into the role of oxidized myelin in microglial function. However, several areas could be improved to strengthen the manuscript. Overall, the study provides valuable insights but requires some additional clarification in the introduction, discussion, and data interpretation. Strengthening the mechanistic explanations and linking findings more directly to MS pathology would enhance the manuscript. Minor adjustments to figures and statistical details would also improve clarity.

Specific Comments:

1. The introduction effectively provides background on MS and microglial activation but could benefit from a clearer statement of the research gap. The transition from general MS pathology to the study rationale needs to be more explicit.
2. The justification for using induced microglia-like cells (iMG) should be expanded, how do these better represent human microglia compared to BV-2 cells?
3. Some references are quite general (e.g., [1,2,3]). Consider including more recent or specialized studies to strengthen the background.
4. The oxidation process is well-described, but the choice of the Cu-peroxide system should be justified further. Are there alternative oxidation methods, and how do they compare?
5. The rationale for using both BV-2 cells and iMGs is clear, but the differences in their microglial behavior should be discussed more explicitly in this section.
6. It would be helpful to specify if corrections for multiple comparisons (e.g., Bonferroni, Tukey) were applied in ANOVA tests.
7. Figure 1 (Oxidation of Myelin): The TBARS assay is an effective way to quantify lipid peroxidation, but were any additional methods (e.g., mass spectrometry, Western blot for oxidized proteins) used to confirm oxidation status?
8. While viability was not affected, could there be more subtle functional consequences of oxidized myelin exposure (e.g., metabolic stress, mitochondrial dysfunction)?
9. The phagocytosis analysis is thorough, but it would be useful to compare the dynamics in BV-2 cells vs. iMGs in greater detail.
10. The absence of a significant increase in pro-inflammatory cytokines is interesting. Could this be due to the experimental time frame? Would a longer exposure to oxidized myelin lead to an inflammatory response?
11. The discussion appropriately highlights the impact of oxidized myelin on oxidative stress and phagocytosis, but it would be beneficial to discuss the implications of these findings in relation to disease progression in MS.
12. The connection between oxidized myelin, autophagy markers (p62), and iron metabolism (Tfr1) is intriguing. Consider expanding on how these pathways might be linked in MS pathology.
13. While the study findings are put into context with existing literature, a direct comparison with other models of oxidized myelin exposure would strengthen the discussion.
14. Figures are generally clear, but some (e.g., phagocytosis data) could be better annotated to highlight key trends. Consider reorganizing the figures so that related results (e.g., oxidative stress markers and phagocytosis data) are closer together in the text.
15. The conclusion is well-written but could better emphasize how these findings might inform future therapeutic strategies. The potential for personalized medicine applications is mentioned but not fully developed, how could these models be used for patient-specific drug screening?
16. Future studies should explore microglia-neuron interactions under oxidized myelin exposure.

Comments on the Quality of English Language

Overall, the manuscript is well-written and generally clear, but there are areas where the language could be improved to enhance readability, clarity, and scientific precision.

Author Response

The manuscript presents an interesting study on microglia-myelin interactions, particularly in the context of oxidative stress and multiple sclerosis (MS). The experimental design is well thought out, and the findings contribute valuable insights into the role of oxidized myelin in microglial function. However, several areas could be improved to strengthen the manuscript. Overall, the study provides valuable insights but requires some additional clarification in the introduction, discussion, and data interpretation. Strengthening the mechanistic explanations and linking findings more directly to MS pathology would enhance the manuscript. Minor adjustments to figures and statistical details would also improve clarity.

Specific Comments:

  1. The introduction effectively provides background on MS and microglial activation but could benefit from a clearer statement of the research gap. The transition from general MS pathology to the study rationale needs to be more explicit.

Response: Thank you for pointing this out. We agree with your comment. Therefore, we have deeply modified this section and add the research gap and the rationale of developing this work.

  1. The justification for using induced microglia-like cells (iMG) should be expanded, how do these better represent human microglia compared to BV-2 cells?

Response: We appreciate your comment. We add three reasons to explain why we use iMG and why they better represent human microglia compared to BV-2 cells specially for MS research in the Introduction.

  1. Some references are quite general (e.g., [1,2,3]). Consider including more recent or specialized studies to strengthen the background.

Response: We agree with this point. We have added several recent references to specialized studies in the revised introduction.

  1. The oxidation process is well-described, but the choice of the Cu-peroxide system should be justified further. Are there alternative oxidation methods, and how do they compare?

Response: We appreciate your insightful comment on this point. We did not find abundant bibliography related to specific oxidation of myelin debris. Cu-peroxide system is cost-effective and reproduce the origin of reactive oxygen species in cells which are hydroxyl radical derived from Fenton reaction. The alternative is Fe-ascorbic system but compared to Cu-peroxide, the effect of the last one is greater oxidizing myelin and destabilizing membranes (doi: 10.1002/jnr.490410209.).

  1. The rationale for using both BV-2 cells and iMGs is clear, but the differences in their microglial behavior should be discussed more explicitly in this section.

Response: Thank you for highlighting this important aspect. We edit the discussion section 3.5 to add this information.

  1. It would be helpful to specify if corrections for multiple comparisons (e.g., Bonferroni, Tukey) were applied in ANOVA tests.

Response: Thank you for pointing this out. We performed Tukey correction test for multiple comparisons. We add this information in the corresponding  Materials and Methods section.

  1. Figure 1 (Oxidation of Myelin): The TBARS assay is an effective way to quantify lipid peroxidation, but were any additional methods (e.g., mass spectrometry, Western blot for oxidized proteins) used to confirm oxidation status?

Response: We are grateful for your thoughtful suggestion. We did not perform any other quantification method of peroxidation. However, we plan to better characterize oxidized myelin with GC-MS analysis in prospective works.

  1. While viability was not affected, could there be more subtle functional consequences of oxidized myelin exposure (e.g., metabolic stress, mitochondrial dysfunction)?

Response: We appreciate your careful review and valuable feedback on this matter. Indeed, we observed autophagy markers and a drop in antioxidant enzymes, suggesting that proteostatic and/or mitochondrial disfunction may be present in a sublethal level.

  1. The phagocytosis analysis is thorough, but it would be useful to compare the dynamics in BV-2 cells vs. iMGs in greater detail.

Response: Thank you for pointing this out. We plan to study this phenomenon in iMG in a large cohort of MS patients. However, we wanted to study first the feasibility of the model as a proof of concept rather than a thorough analysis in iMG.

  1. The absence of a significant increase in pro-inflammatory cytokines is interesting. Could this be due to the experimental time frame? Would a longer exposure to oxidized myelin lead to an inflammatory response?

Response: Thank you for your thoughtful observation. We don’t think it is a matter of time since previous experiments reported an efficient response in 24 hours at least. We rather think that it is a matter of the stimuli which might not activate the transcription factors associated with inflammatory cytokines.  doi: 10.1007/s12192-014-0552-1

  1. The discussion appropriately highlights the impact of oxidized myelin on oxidative stress and phagocytosis, but it would be beneficial to discuss the implications of these findings in relation to disease progression in MS.

Response: We sincerely appreciate your careful consideration of this issue. Herein, we add information regarding this important feature in the section 3.4 of the revised manuscript.

  1. The connection between oxidized myelin, autophagy markers (p62), and iron metabolism (Tfr1) is intriguing. Consider expanding on how these pathways might be linked in MS pathology.

Response: Thank you for your consideration. We add this information in 3.3 section of the revised manuscript.

  1. While the study findings are put into context with existing literature, a direct comparison with other models of oxidized myelin exposure would strengthen the discussion.

Response: We appreciate your insightful comment. Therefore, we add this comparison in section 3.2.

  1. Figures are generally clear, but some (e.g., phagocytosis data) could be better annotated to highlight key trends. Consider reorganizing the figures so that related results (e.g., oxidative stress markers and phagocytosis data) are closer together in the text.

Response: We fully agree with this point. Therefore, we change the 2.3 section of pro-inflammatory cytokines to 2.5 section, so now the oxidative stress markers and phagocytosis data are closer in the text. We also changed the description in the Figure Legends of the figures related to phagocytosis data to better describe the content.  

  1. The conclusion is well-written but could better emphasize how these findings might inform future therapeutic strategies. The potential for personalized medicine applications is mentioned but not fully developed, how could these models be used for patient-specific drug screening?

Response: Thank you for highlighting this important prospective point of view. Herein, we address this question in more detail in the section 3.6 of the revised manuscript.

  1. Future studies should explore microglia-neuron interactions under oxidized myelin exposure.

Response: We appreciate this recommendation. Therefore, we aim to set up mixed culture of hiPSC derived neurons and iMG to tackle potential interactions in a more comprehensive cell model.

Comments on the Quality of English Language

Overall, the manuscript is well-written and generally clear, but there are areas where the language could be improved to enhance readability, clarity, and scientific precision.

Round 2

Reviewer 2 Report

Comments and Suggestions for Authors

The authors have addressed all my prior concerns thoroughly. However, the study demonstrates changes in Tfr1 and autophagy markers (p62) in response to oxidized myelin. Since iron accumulation is a key factor in MS progression, ferroptosis may play a significant role. The authors could expand their discussion on ferroptosis as a potential mechanism of neurotoxicity in MS, particularly in the context of microglial dysfunction. Further validation of lipid peroxidation markers could be beneficial. 

Figures 4 & 5 (phagocytosis and fluorescence quantifications) are dense and difficult to interpret. Use consistent color coding for different treatments and improve figure legends for better readability.

The manuscript uses terms like "CTL myelin," "control myelin," and "non-oxidized myelin" interchangeably. Maintain consistent terminology throughout the text to improve readability.

Author Response

Comments and Suggestions for Authors

The authors have addressed all my prior concerns thoroughly. However, the study demonstrates changes in Tfr1 and autophagy markers (p62) in response to oxidized myelin. Since iron accumulation is a key factor in MS progression, ferroptosis may play a significant role. The authors could expand their discussion on ferroptosis as a potential mechanism of neurotoxicity in MS, particularly in the context of microglial dysfunction. Further validation of lipid peroxidation markers could be beneficial.

-Response: Thank you for your insightful comment. We agree with you so we have expanded the 3.3 section according to your comments.

Figures 4 & 5 (phagocytosis and fluorescence quantifications) are dense and difficult to interpret. Use consistent color coding for different treatments and improve figure legends for better readability.

-Response: We appreciate your suggestions. We have changed the colors to enhance the contrast between the treatments in all figures and add a brief description of the results in the Figure Legends of 4 and 5.

The manuscript uses terms like "CTL myelin," "control myelin," and "non-oxidized myelin" interchangeably. Maintain consistent terminology throughout the text to improve readability.

-Response: Thank you for your observations. We have changed “control” and “non-oxidized” to CTL, so now it matches also with the terminology in the Figures.